# What Drives Animal Fluency Performance in Cantonese-Speaking Chinese Patients with Adult-Onset Psychosis?

**DOI:** 10.3390/brainsci13030372

**Published:** 2023-02-21

**Authors:** Christy Lai-Ming Hui, Sally Hiu-Wah See, Tsz-Ching Chiu, Andrea Stephanie Pintos, Johanna M. Kroyer, Yi-Nam Suen, Edwin Ho-Ming Lee, Sherry Kit-Wa Chan, Wing-Chung Chang, Brita Elvevåg, Eric Yu-Hai Chen

**Affiliations:** 1Department of Psychiatry, School of Clinical Medicine, University of Hong Kong, Hong Kong, China; 2State Key Laboratory of Brain and Cognitive Sciences, University of Hong Kong, Hong Kong, China; 3Department of Clinical Medicine, University of Tromsø—The Arctic University of Norway, 9037 Tromsø, Norway

**Keywords:** psychosis, schizophrenia, semantic, category fluency, chinese zodiac signs

## Abstract

Among the numerous studies investigating semantic factors associated with functioning in psychotic patients, most have been conducted on western populations. By contrast, the current cross-sectional study involved native Cantonese-speaking Chinese participants. Using the category fluency task, we compared performance between patients and healthy participants and examined clinical and sociodemographic correlates. First-episode psychosis patients (*n* = 356) and gender- and age-matched healthy participants (*n* = 35) were asked to generate as many ‘animals’ as they could in a minute. As expected, patients generated fewer correct responses (an average of 15.5 vs. 22.9 words), generated fewer clusters (an average of 3.7 vs. 5.4 thematically grouped nouns), switched less between clusters (on average 8.0 vs. 11.9 switches) and, interestingly, produced a larger percentage of Chinese zodiac animals than healthy participants (an average of 37.7 vs. 24.2). However, these significant group differences in the clusters and switches disappeared when the overall word production was controlled for. Within patients, education was the strongest predictor of category fluency performance (namely the number of correct responses, clusters, and switches). The findings suggest that an overall slowness in patients may account for the group differences in category fluency performance rather than any specific abnormality *per se*.

## 1. Introduction

Neurocognitive dysfunction is considered to be a core deficit in patients with psychotic disorders including schizophrenia [1,2]. It has been identified as a strong predictor of social and functional outcomes in patients with schizophrenia [3,4], and commonly presents as a variety of deficits in sustained and selective attention [5,6], working memory [6,7], executive function [6,8], verbal memory [9,10], and semantic memory [11,12]. Semantic memory has been of major interest due to its potential relationship to some psychotic symptoms, notably formal thought disorder [13]. Semantic memory represents the basic vocabulary and general knowledge about the world, namely the facts and experiences accumulated in an individual’s lifetime [14]. Theories about semantics, namely the knowledge of objects and concepts, have proposed a network organization where associated concepts are those that share more common attributes, and are more strongly linked than those sharing fewer attributes [15].

Numerous experimental paradigms have been leveraged to probe the integrity of this semantic system, notably semantic priming, speeded categorization, and category fluency. In the category fluency task, participants are required to generate as many exemplars as possible from a specific category within a timed period (e.g., naming as many ‘animals’ as possible within one minute) [16] and performance scores are generated based on the number of correct and incorrect words [17], and measures of semantic clustering and switching can be computed on this output [18]. Previous studies that have adopted this task and method of analysis have not surprisingly found that schizophrenia patients generate fewer correct responses and show less semantic clustering than healthy controls [19,20,21,22,23], and have concluded that these differences are related to strategic memory retrieval problems (rather than a degraded memory store *per se*) [24], but the role of antipsychotic medication in such findings is not clear [25]. Other studies have suggested somewhat similarly that the poorer performance is due to ‘difficulties finding new words within a semantic field’ attributable to general slowness [22], or that it is due to problems in inhibiting irrelevant information [26]. In total, this has been taken to suggest that patients with schizophrenia have disproportionate difficulties that emerge in semantic processing as compared to healthy participants. Further, it has been shown that more severe negative symptoms are associated with fewer correct responses in the category fluency task in elderly schizophrenia patients [27]. However, in terms of positive symptoms, such as formal thought disorder, findings from the category fluency task have been inconsistent and—according to a recent meta-analysis—there are too few studies to form a meaningful conclusion [13].

When generating words in the category fluency task, the words we choose are largely culturally and geographically determined. Consider the simple example of generating as many ‘animals’ as one can in a minute. The actual animals will likely include ‘panda’ for those living in China since this animal is an iconic symbol of the country. Furthermore, in Chinese culture, the zodiac astrology is a traditional way to calculate the days in the calendar that features 12 animal signs rotating each year in the order of *rat*, *ox*, *tiger*, *rabbit*, *dragon*, *snake*, *horse*, *sheep*, *monkey*, *rooster*, *dog,* and *pig*. Each person’s year of birth is associated with a zodiac animal, and it is believed that these signs determine important aspects regarding personalities, careers, marriage, and fortune. Since there is considerable significance assigned to these zodiac animal signs in Chinese culture, we would expect to see these cultural differences emerge when generating lists of animals, and thus it would be interesting to formally examine whether the generation of animals specifically is affected by the cultural importance of the zodiac animals and whether this is differentially affected by psychosis. Therefore, in order to examine these issues, this study involved only Cantonese-speaking Chinese patients with *adult-onset* psychosis in which the task was to name as many animals as possible within one minute:

(1) We compared patients and healthy controls in terms of fluency performance (as defined by the numbers of correct, repeated and incorrect responses, clusters, and switches) as well as the percentage of Chinese zodiac animals reported with and without controlling the potential confounder of overall word production; and

(2) We explored the clinical and sociodemographic correlates of fluency performance in patients.

## 2. Materials and Methods

### 2.1. Participants

Three-hundred and sixty Cantonese-speaking Chinese outpatients meeting DSM-IV criteria for schizophrenia, schizoaffective disorder, schizophreniform disorder, brief psychotic disorder, psychosis not otherwise specified, or delusional disorder were recruited from the Jockey Club Early Psychosis (JCEP) Project in Hong Kong. The project provided a territory-wide early intervention service to those with first-episode psychosis aged between 26 and 55 years [28,29]. Patients were recruited within a 4-month time interval following their first contact with the psychiatric service. Exclusion criteria for all potential participants were: organic brain conditions, known history of intellectual disability, or risk of suicide or violence. Case medical officers confirmed whether the patients met the inclusion or exclusion criteria. Finally, after excluding 4 participants with missing category fluency data, there was a total of 356 participants in the current study. Thirty-five healthy adults from the Department of Psychiatry, University of Hong Kong were recruited to act as control participants. They were aged between 18 to 55 years and prior to participation were screened for a family history of psychiatric disorders or a medical history of antipsychotic usage. The study was approved by the Institutional Review Boards and was conducted in accordance with Good Clinical Practice and with the Declaration of Helsinki. All patients and controls provided written informed consent. 

### 2.2. Materials

All assessments were conducted at study entry through face-to-face interviews by trained research assistants. All participants were assessed for their basic demographics and were administered the category fluency task. For patients, estimates of premorbid functioning, clinical demographics, functioning, and positive and negative symptoms were also assessed.

#### 2.2.1. Basic Demographics, Premorbid Functioning, and Symptoms

Basic demographic information including age, gender, years of education, marital status, employment status, and age of onset were recorded. Estimates of premorbid functioning during childhood, adolescence, and adulthood were evaluated using the Premorbid Adjustment scale (PAS) [30]. Premorbid schizoid and schizotypal traits, including affect, suspiciousness, antisocial behavior, asocial behavior, and other abnormalities, were assessed using the Assessment of Premorbid Schizoid and Schizotypal Traits (PSST) [31]. Duration of untreated psychosis (DUP), defined as the period between the onset of positive psychotic symptoms and first contact with any psychiatric service, was assessed using the Interview for the Retrospective Assessment of the Onset of Schizophrenia (IRAOS) [32]. It is a standardized, semi-structured interview conducted on patients and their close relatives. Positive and negative symptoms were assessed using the Positive and Negative Syndrome Scale (PANSS) [33], the Scale for the Assessment of Positive Symptoms (SAPS) [34], and the Scale for the Assessment of Negative Symptoms (SANS) [35].

#### 2.2.2. Category Fluency Task

Participants were asked to produce as many animals as possible within one minute. The responses were voice-recorded, transcribed into Chinese, then translated into English for coding purposes. The task performance for category fluency was scored using a qualitative method [18], which included (i) counting the total number of correct words generated, (ii) calculating the percentage of animals reported that were Chinese zodiac signs, (iii) computing the number of clusters generated, and (iv) the number of switches between clusters. In addition, we scored the (v) total number of incorrect animals, and (vi) the total number of repeats. The scoring method used to compute clusters and switches (defined below) differed slightly from Troyer’s original method, in which repeated and incorrect responses were excluded from all calculated scores.

Clustering was defined as groups of contiguous words belonging to the same semantic subcategories, which are organized by zoological categories (i.e., canine, reptile), human use (i.e., farm animals) and living environment (i.e., in Africa, on a farm). For instance, if a patient generated the sequence “pig, sheep, dog”, then “pig” and “sheep” would be classified within the farm animal cluster, and “dog” in the pet cluster. Therefore, the total number of clusters generated here would be two (pet and farm animals). In the case that an animal could be classified into two subcategories, it would be counted towards both clusters. In addition, if a smaller cluster was contained within a larger cluster, only the larger one would be adopted.

Switching was calculated as the total number of transitions between clusters, including single words. In a case where there were three sub-categorical clusters (farm, African, and pet), transitioning from the farm animal cluster to the African animal cluster would be counted as one, and transitioning further from the African animal cluster to the pet cluster would be counted as two.

We also computed the percentage of Chinese zodiac animals reported. This was calculated by dividing the total number of zodiac animals (i.e., rat, cow, tiger, rabbit, dragon, snake, horse, sheep, monkey, chicken, dog, and pig) by the total number of correct animals reported and multiplying by 100. A higher percentage represented a greater number of zodiac signs named. It was noted that while ‘dragon’ was considered an incorrect response, it was still counted as part of the zodiac percentage as ‘dragon’ is one of the twelve Chinese zodiac animals.

Finally, in order to examine whether the overall word production affected any group differences in the fluency parameters listed above, word production—as defined by the total number of responses (i.e., including correct, incorrect, and repeated responses) divided by 60 s—was calculated.

### 2.3. Data Analysis

Data were analyzed using IBM Statistical Package for Social Science (SPSS) version 25.0. Differences in basic demographics and category fluency performance (defined as correct, incorrect, and repeated responses, number of clusters, number of switches, and percentage of Chinese zodiac animals) between patients and healthy participants were assessed using independent samples t-tests and Chi-squared tests. Linear regression models were conducted to examine the confounding effects of word production and years of education, separately, on the differences in these category fluency parameters between patients and healthy participants.

Pearson correlations were conducted within the patient group to explore relationships between category fluency performance and possible correlates including basic demographics, functioning as well as clinical symptoms. To control the false discovery rate (FDR) due to multiple comparisons, the threshold of the q-value was set at 0.1. Subsequently, only correlates with a *p*-value of ≤ 0.05 remained after the FDR correction was included in the subsequent multiple linear regression model. Moreover, when significant univariate variables were highly correlated (e.g., PANSS negative symptoms subscale and SANS measures), only one factor was selected for inclusion. We ran four multiple linear regression models separately for each item that reflects category fluency performance, namely (i) correct responses, (ii) the number of clusters, (iii) the number of switches, and (iv) the percentage of Chinese zodiac signs. We did not run multiple regression models for the remaining two performance measures (“repeated” and “incorrect responses”) of the category fluency task, as none-to-minimal associated factors were identified from the univariate correlational analyses.

## 3. Results

### 3.1. Participants’ Characteristics

Of the 360 first-episode psychosis patients, 4 (1.1%) were excluded due to missing category fluency data (thus resulting in a sample of 356 patients). Table 1 shows that the average age of patients was 38.2 years while that of healthy participants was 34.4 years. The proportion of males in the patient group (43.5%) and the healthy group (45.7%) was not significantly different. However, not surprisingly, healthy participants received significantly more years of education (*M* = 15.5, *SD* = 3.0) than patients did (*M* = 10.7, *SD* = 3.8), *p* < 0.001.

Within the patient group, the average age of illness onset was 36.5 years, and the median DUP was approximately three months. Around forty percent of the patients were diagnosed with schizophrenia (43.5%), with an average PANSS total of 42.4 at study entry. The average antipsychotic medication dosage at baseline (defined by chlorpromazine equivalents [CPZe]) was 168.6. About two-thirds of the patients (70.8%) received second- generation antipsychotics.

### 3.2. Category Fluency Performance between Patients and Healthy Participants

As expected, compared to healthy participants, patients generated significantly fewer correct responses on the category fluency task, and there were fewer clusters and switches between clusters (all *p* < 0.001) (Table 1). Interestingly, of the animal words generated, patients produced a significantly larger percentage of Chinese zodiac animals than healthy participants (*p* < 0.001). The number of incorrect responses and repeated responses did not differ between the two groups (*p* > 0.05). Notably, all the aforementioned results remained even after education was controlled for, thus excluding education as the reason for the observed group differences.

To establish whether the observed group differences in animal fluency were artifacts of patients’ poorer overall word production, the above analyses were repeated with the word count taken into consideration. Table 1 shows that the significant group differences in the number of clusters and switches disappeared after controlling for the total number of words produced over 60 s, thus implying that overall word production has a potential role in mediating the relationship between the number of clusters and switches in the two groups.

### 3.3. Correlations between Category Fluency and Basic Demographics in Patients

An older age of onset was associated with poorer animal fluency performance (Table 2), specifically fewer correct responses (*p* < 0.001), more repetitions (*p* = 0.007), fewer clusters (*p* = 0.004), less switching (*p* < 0.001), and a higher percentage of Chinese zodiac signs (*p* < 0.001). Less education correlated with fewer correct responses (*p* < 0.001), fewer clusters (*p* < 0.001), less switching (*p* < 0.001), and a higher percentage of zodiac animals (*p* < 0.001). Interestingly, female patients generated a higher percentage of zodiac animals than male patients (*p* = 0.003). Poorer social and occupational functioning in patients was associated with fewer correct responses (*p* < 0.001), fewer clusters (*p* = 0.002), fewer switches (*p* = 0.002), and a higher percentage of zodiac animals (*p* = 0.005). Finally, there were no significant differences between any of the category fluency measures and the dosage of medication intake, premorbid adjustment, DUP, as well as diagnosis (schizophrenia versus non-schizophrenia) (all *ps* > 0.05).

### 3.4. Correlations between Category Fluency and Symptoms in Patients

Not surprisingly, patients who were more impaired clinically were also those who performed poorer on the animal fluency task (Table 2). Patients with more severe negative and general psychopathology symptoms as well as a higher total scores on the PANSS produced fewer correct words (all *p* < 0.001) and fewer clusters (all *p* < 0.01). Those with more severe negative symptoms, general psychopathology symptoms, and a higher score on the PANSS also performed fewer switches (*p* = 0.003; *p* < 0.001; *p* < 0.001, respectively). Looking closer at negative symptoms, more severe affective blunting, alogia, anhedonia-asociality, and total SANS scores were associated with fewer correct responses (all *p* < 0.01) while more severe alogia and total SANS scores were also associated with a fewer number of clusters (all *p* < 0.01). Meanwhile, patients who scored higher on affective blunting, alogia, avolition-apathy, anhedonia-asociality, and had a higher total SANS score switched less between clusters (all *p* < 0.01). There were no significant correlations between positive symptoms of the SAPS and animal fluency performance (all *p* > 0.05), except that inappropriate affect negatively correlated with the percentage of zodiac animals (*p* = 0.004). In summary, the patients who performed more poorly on the animal fluency task were those with more severe negative symptoms, general psychopathology symptoms, and a higher total PANSS score.

### 3.5. Multiple Linear Regression Analyses on Factors Related to Category Fluency Performance in Patients

In order to assess whether the aforementioned factors with significant correlation coefficients were predictors of animal fluency performance, we performed multiple linear regression analyses. Regression coefficients are displayed in Table 3. 

Significant factors associated with the number of correct responses in the animal fluency task (i.e., age of onset, years of education, social, and occupational functioning, general psychopathology, blunting, alogia, anhedonia, and asociality) were entered into a multiple linear regression model. Only years of education significantly predicted the number of correct animal responses (*β* = 0.507, *p* < 0.001). The results indicated that the model explained 21% of the variance.

Similarly, among the five significant correlates for the number of clusters in the fluency task (i.e., age of onset, years of education, social and occupational functioning, general psychopathology, alogia), only years of education was a significant predictor (*β* = 0.129, *p* < 0.001). The model explained about 16% of the variance.

Of the eight significant correlates for the number of switches in the category fluency task (i.e., age of onset, years of education, social, and occupational functioning, general psychopathology, blunting, alogia, avolition and apathy, anhedonia, and asociality), younger age of onset (*β* = −0.050, *p* = 0.013) and more years of education (*β* = 0.211, *p* < 0.001) significantly predicted more switches. Together, the model explained about 16% of the variance.

Finally, based on the significant factors (i.e., age of onset, years of education, gender, social, and occupational functioning, or inappropriate affect), a multiple regression analysis showed that fewer years of education (*β* = −0.985, *p* < 0.001), female (*β* = 3.930, *p* = 0.022), poorer social and occupational functioning (*β* = −0.172, *p* = 0.012), and less severe positive symptoms regarding inappropriate affect (*β* = −5.933, *p* = 0.008) significantly predicted the higher percentage of zodiac animals named. Together, the model explained about 15% of the variance.

In sum, the number of years of education has a prominent effect on animal fluency performance in patients with psychosis, as it was consistently identified as a significant factor associated in multiple regression models. Specifically, less education was associated with fewer correct responses, fewer clusters, less switching, and a greater percentage of zodiac animals being generated.

## 4. Discussion

Among studies that have investigated semantic functioning and the relationship with demographics and clinical symptoms, this is one of the few studies involving only Cantonese-speaking, adult-onset Chinese psychosis patients in Hong Kong. As expected, patients produced fewer correct responses on the animal fluency task, and these responses revealed fewer clusters and less switching as compared to healthy participants, which is consistent with previous studies [19,20,21,22,23]. However, when overall word production was controlled for, the group difference in the number of clusters and switches disappeared, thus implying that the overall word production potentially accounted for the group differences. As patients produced fewer words overall, this further suggests that general slowness may be driving some of the observed qualitative differences in patients: a general slowness results in fewer words being produced, which thus affords less possibility of generating many clusters and thus switching less between these clusters. The importance of general slowness and psychomotor speed in such tasks in schizophrenia patients echoes previous studies [22,36]. There are of course many possible reasons for general slowing in patients, with the most likely candidates being ill in general and being ill with psychosis specifically [37,38]. Although our study did not examine the cognitive or pharmacological mechanisms for slowing in patients, the findings suggest that overall slowness could account for group differences in naming animal words in a category fluency task.

This study found that education had a prominent effect on animal fluency performance in this cohort of patients. Importantly, it was the only variable that remained significant in the multiple regression models, predicting the number of correct responses and cluster numbers. It was also one of the two significant factors predicting the number of switches. Our univariate correlation analysis found that more severe negative symptoms, including blunt affect, alogia, avolition-apathy, and anhedonia-asociality, were associated with fewer correct responses and less switching in the animal fluency task. Previously, it has been reported that schizophrenia patients with negative symptoms generate fewer words than those without negative symptoms [24]. In the current study, we did not find any significant correlations between formal thought disorder (and also other positive symptoms) and fluency performance. In terms of demographics, we found that an older age of onset correlated with poorer performance, namely fewer correct responses, more repeated items, and less clustering and switching. A possible explanation is that getting old is associated with general slowing [39,40,41,42]. As the category fluency task is a speeded task, arguably just being slower means one is more likely to generate fewer words, potentially be forgetful and thus repeat a bit, and so with less word generation there is less scope for clustering and switching. Our findings contradict a previous study where no significant associations were observed between the age of onset and category fluency performance [43]. Finally, no significant relationships were observed between category fluency performance and gender, DUP, or specific diagnosis.

Within our sample, a higher proportion of patients reported Chinese zodiac signs in the animal fluency task than healthy participants. In doing the task, participants were not told to avoid generating any words from the Chinese zodiac signs and thus the task is one of how participants individually interpret the task instructions. While we did not investigate the reason for such a group difference, we explored in patients the clinical and functional correlates of producing animals that were also zodiac signs. Interestingly, patients with less education had poorer social and occupational functioning, presented with less severe inappropriate affect, and those who were female were more likely to name zodiac animal signs in the fluency task. We conducted a further analysis (not reported in the Results) and found that females and males performed similarly in terms of correct responses (average = 15.4 in females, average = 15.5 in males, *p* = 0.884), which implies that female patients reporting more zodiac animals still holds true when overall performance is accounted for. Moreover, our results suggest that generating these zodiac animals may not necessarily be effective in terms of boosting overall performance, because a higher percentage of Chinese zodiac signs was, in fact, associated with fewer correct responses, more incorrect and repeated responses, as well as fewer clusters and switches.

Despite the category fluency task being relatively simple to administer, its scoring methods are not without limitations. The determination of cluster boundaries relies on subjective judgment [44] and the coding of these markers can be time-consuming. To go beyond manual simple word counts, new analytic methods have been proposed to collect fluency data digitally and analyze the data using automatic methods that capture the exact timing between utterances and thus the temporal dynamics of speech [45]. Finally, there has been discussion concerning the extent to which this data elicited by the category fluency task can be thought to assay semantics, given the derived similarities are not stable across time points in healthy participants [46], and in patients, the variance can be attributed to many non-semantic factors [44,47].

## 5. Conclusions

This was one of the first studies on language and semantics that was conducted in 356 Cantonese-speaking psychosis patients with an onset after 25 years old in Hong Kong. Using the animal fluency task, we found that patients produced fewer correct responses, generated fewer clusters, and switched less between these clusters than healthy participants, and that the groups produced an equal number of incorrect responses. This result was unrelated to the role of education but partly attributed to the overall word production—because the significant group differences for clusters and switches disappeared after overall word production was controlled for. Since patients produced fewer words than controls, this further suggests that an overall slowness in patients may be driving some of the observed differences in patients, rather than an abnormality *per se*.

We specifically explored the clinical and sociodemographic correlates of these fluency performances in patients and found a close association with educational attainment and the presence of negative symptoms. Finally, although this study was not designed to tease apart the underlying cognitive or pharmacological mechanisms underlying task performance, the tantalizing cultural findings in the patient group do lay the foundation for a future study that is designed specifically to examine how these mechanisms in language may be affected by a combination of psychosis and cultural factors.

## Figures and Tables

**Table 1 brainsci-13-00372-t001:** Basic characteristics and category fluency performance in first episode psychosis patients (*n* = 365) and healthy participants (*n* = 35).

Characteristics		Patients (*n* = 356)	Healthy Participants (*n* = 35)	*p*-Value	*p*-Value *	*p*-Value *^#^*
Age at testing, average (SD)		38.2 (8.4)	34.4 (11.1)	0.056	/	/
Age of illness onset, average (SD)		36.5 (8.7)	/	/	/	/
Male, *n* (%)		155 (43.5)	16 (45.7)	0.805	/	/
Years of education, average (SD)		10.7 (3.8)	15.5 (3.0)	<0.001	/	/
DUP, days, median (IQR)		92.5 (20.0–385.3)	/	/	/	/
Diagnosis, *n* (%)			/	/	/	/
	Schizophrenia	155 (43.5)				
	Delusional disorder	71 (19.9)				
	Schizophreniform disorder	60 (16.9)				
	Brief psychotic disorder	42 (11.8)				
	Psychosis not otherwise specified	20 (5.6)				
	Manic episodes with psychotic features	5 (1.4)				
	Schizoaffective disorder	3 (0.8)				
Medication intake, CPZe, average (SD)		168.6 (144.6)	/	/	/	/
Medication type, *n* (%)			/	/	/	/
First generation antipsychotics Second generation antipsychotics Mixed No medication		88 (24.7) 252 (70.8) 4 (1.1) 12 (3.4)				
PAS, average (SD)		0.2 (0.2)	/	/	/	/
PANSS, average (SD)			/	/	/	/
	Total PANSS	42.4 (12.1)				
	Positive symptoms	9.1 (3.6)				
	Negative symptoms	10.2 (4.4)				
	General psychopathology	23.0 (7.2)				
SOFAS, average (SD)		54.5 (12.5)	/	/	/	/
Category fluency measures, average (SD)						
	Correct response	15.5 (5.5)	22.9 (4.4)	<0.001	<0.001	<.001
	Incorrect response	0.2 (0.5)	0.2 (0.5)	0.813	0.783	0.670
	Repeated response	1.1 (1.8)	0.9 (1.0)	0.630	<0.001	0.645
	Number of clusters	3.7 (1.6)	5.4 (1.5)	<0.001	0.114	<0.001
	Number of switches	8.0 (3.3)	11.9 (3.1)	<0.001	0.090	<0.001
	Percentage of Chinese zodiac signs	37.7 (16.8)	24.2 (10.7)	<0.001	0.003	0.012
	Overall word production	0.3 (0.1)	0.4 (0.1)	<0.001	/	<0.001

* *p*-values after controlling for overall word (animal) production; calculated by the total number of responses (correct, incorrect, and repeated) divided by 60 s. ^#^
*p*-values after controlling for years of education. Abbreviations: CPZe = Chlorpromazine equivalents; DUP = duration of untreated psychosis; IQR = interquartile range; PAS = premorbid adjustment scale; PSST = premorbid schizoid and schizotypal traits; PANSS = positive and negative syndrome scale; SD = standard deviation; SOFAS = social and occupational functioning assessment scale.

**Table 2 brainsci-13-00372-t002:** Correlation coefficients between clinical assessment and animal fluency performance in patients.

		Correct Response	Incorrect Response	Repeated Response	Number of Clusters	Number of Switches	Percentage of Chinese Zodiac Signs
		r/t	r/t	r/t	r/t	r/t	r/t
Age of onset		**−0.195 *****	0.027	**0.143 ****	**−0.153 ****	**−0.223 *****	**0.214 *****
Gender		0.146	−1.305	−0.779	0.786	−0.963	**−2.977 ****
Years of education		**0.409 *****	−0.013	0.008	**0.361 *****	**0.322 *****	**−0.304 *****
Diagnosis		−0.144	1.309	−0.116	−0.522	−0.802	−0.233
Medication intake (CPZe)		−0.044	−0.027	0.068	−0.057	−0.039	0.042
PAS		−0.041	0.004	−0.049	−0.004	−0.037	0.052
DUP		−0.083	−0.061	−0.111 *	−0.067	−0.078	0.021
SOFAS		**0.191 *****	−0.036	−0.015	**0.166 ****	**0.164 ****	**−0.147 ****
PANSS							
	Total PANSS	**−0.186 *****	0.004	−0.045	**−0.149 ****	**−0.184 *****	0.059
	Positive symptoms	−0.013	0.039	−0.018	−0.002	−0.025	−0.054
	Negative symptoms	**−0.186 *****	0.024	0.009	**−0.149 ****	**−0.159 ****	0.087
	General psychopathology	**−0.191 *****	−0.028	−0.073	**−0.158 ****	**−0.199 *****	0.073
SAPS							
	Total SAPS	−0.005	0.028	−0.028	−0.004	−0.014	−0.069
	Hallucination	0.008	0.023	−0.055	0.003	−0.017	−0.031
	Delusion	0.015	0.066	−0.007	0.013	0.021	−0.040
	Bizarre behavior	−0.045	−0.003	0.034	−0.058	−0.005	−0.021
	Formal thought disorder	−0.034	−0.050	−0.014	−0.011	−0.057	−0.078
	Inappropriate affect	0.018	−0.055	−0.029	0.008	−0.003	**−0.152 ****
SANS							
	Total SANS	**−0.202 *****	0.028	−0.007	**−0.146 ****	**−0.219 *****	0.080
	Blunt affect	**−0.164 ****	0.028	−0.016	−0.099	**−0.172 ****	0.026
	Alogia	**−0.209 *****	0.051	−0.038	**−0.161 ****	**−0.213 *****	−0.016
	Avolition−apathy	−0.122 *	0.043	0.009	−0.119 *	**−0.158 ****	0.108 *
	Anhedonia−asociality	**−0.148 ****	0.018	0.005	−0.116 *	**−0.156 ****	0.124 *
	Attention	−0.096	−0.085	0.017	−0.029	−0.110 *	−0.022

* *p* < 0.05, ** *p* < 0.01, *** *p* < 0.001; significant *p*-values after FDR correction are in bold. Abbreviations: CPZe = Chlorpromazine equivalents; DUP = duration of untreated psychosis; IQR = interquartile range; PAS = premorbid adjustment scale; PSST = premorbid schizoid and schizotypal traits; PANSS = positive and negative syndrome scale; SAPS = Scale for the Assessment of Positive Symptoms; SANS = Scale for the Assessment of Negative Symptoms; SD = standard deviation; SOFAS = social and occupational functioning assessment scale.

**Table 3 brainsci-13-00372-t003:** Multiple linear regression analyses of factors related to category fluency performance in patients.

Predictors	Correct Responses (*n* = 356)				Number of Clusters (*n* = 356)				Number of Switches (*n* = 356)				Percentage of Chinese Zodiac Signs (*n* = 356)			
	*β*	SE	*p*	95% CI	*β*	SE	*p*	95% CI	*β*	SE	*p*	95% CI	*β*	SE	*p*	95% CI
Age of onset	−0.043	0.032	0.188	−0.106 to 0.021	−0.006	0.009	0.494	−0.025 to 0.012	−0.050	0.020	**0.013** *	−0.089 to −0.010	0.200	0.103	0.052	−0.002 to 0.402
Education years	0.507	0.075	**<0.001 *****	0.359 to 0.654	0.129	0.022	**<0.001 *****	0.087 to 0.172	0.211	0.047	**<0.001 *****	0.120 to 0.303	−0.985	0.236	**<.001 *****	−1.450 to −0.521
Gender	n/a				n/a				n/a				3.930	1.708	**0.022 ***	0.571 to 7.289
SOFAS	0.035	0.024	0.147	−0.012 to 0.083	0.009	0.007	0.164	−0.004 to 0.023	0.013	0.015	0.388	−0.017 to 0.043	−0.172	0.068	**0.012 ***	−0.307 to −0.037
PANSS general psychopathology	−0.040	0.045	0.384	−0.129 to 0.050	−0.010	0.012	0.407	−0.034 to 0.014	−0.032	0.030	0.282	−0.090 to 0.026	n/a			
SAPS inappropriate affect	n/a				n/a				n/a				−5.933	2.230	**0.008 ****	−10.319 to 1.546
SANS blunting	−0.040	0.064	0.527	−0.165 to 0.085	n/a				−0.023	0.040	0.574	−0.102 to 0.056	n/a			
SANS alogia	−0.247	0.139	0.078	−0.521 to 0.028	−0.060	0.032	0.066	−0.124 to 0.004	−0.164	0.087	0.060	−0.334 to 0.007	n/a			
SANS avolition apathy	n/a				n/a				−0.012	0.063	0.844	−0.136 to 0.111	n/a			
SANS anhedonia asociality	0.007	0.062	0.914	−0.116 to 0.129	n/a			—	−0.001	0.040	0.981	−0.080 to 0.078	n/a			
	**F**	** *R* ^2^ **	** *p* **		**F**	** *R* ^2^ **	** *p* **		**F**	** *R* ^2^ **	** *p* **		**F**	** *R* ^2^ **	** *p* **	
**Overall Model**	**13.259**	0.211	**<0.001 *****		12.881	0.155	**<0.001 *****		**8.512**	0.164	**<0.001 *****		**12.057**	0.147	**<0.001 *****	

* *p* < 0.05, ** *p* < 0.01, *** *p* < 0.001; significant *p* values are in bold. Abbreviations: PANSS = positive and negative syndrome scale; SAPS = Scale for the Assessment of Positive Symptoms; SANS = Scale for the Assessment of Negative Symptoms; SOFAS = social and occupational functioning assessment scale.

## Data Availability

The data that support the findings of this study are available upon reasonable request.

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
