# Peer review of "What Drives Animal Fluency Performance in Cantonese-Speaking Chinese Patients with Adult-Onset Psychosis?"

_brainsci, 2023, doi:10.3390/brainsci13030372_

Round 1
Reviewer 1 Report
The work fills the gap in cross-linguistic work on the functioning of the semantic factors in psychotic patients by examining the performance of the Cantonese-speaking Chinese patients on a semantic category fluency task. I can recommend this work for the publication as part of the special issue on the cognitive neuroscience approaches to the psychopathology of psychotic disorders after the authors flash out the connection of their results to the predicted mechanisms that result in the task performance deficiency in the patients.
More specifically:
Introduction:
p2. Line 45 Could you explain to what mechanisms Elvevag and colleagues (2002) attribute “general slowness” in schizophrenia patients and how the mechanisms that they hypothesis are different from an alternative mechanism that results in “retrieval deficits” hypothesized by Allen and colleagues (1993).
Such explanation is needed to understand better the contribution of this study, which also needs to come clearer in the Discussion session
p12. Lines 319-320 – please explain why do you think the older age of onset correlated with poorer performance such as fewer correct responses more repeated items, and less clustering and switching.
P12. Lines 359-360 – Same here – Why do you think patients are slower in naming – what mechanisms do you think your results support better?

Reviewer 2 Report
The paper is quite solid. This Reviewer does not know to what extent it can be considered relevant, but it is surely a good effort, also written adequately, up to academic standards.
The sample of participants is relatively large, and that is good, because, at least theoretically, it allows the Authors to get indicative (or relatively indicative) results.
The Introduction is ok, quite effective.
After the Introduction, in place of the Materials and Methods section, a proper Literature Review section would be more appropriate, focused on the works used and cited in this study and with a broader outline of the research in the field, also a little more 'general', which would make the paper more comprehensive, and even more 'user-friendly', especially for a non-specialized 'audience'.
Then, after that, a proper Methodology section would be welcomed.
The suggestion is, therefore, to split the current second section into two parts, one dedicated to the Literature Review and the other to the Methodology.
The Results section is ok, perhaps it could be formatted a little better and it could include some additional 'preliminary comments' anticipating the Discussion.
The Discussion should be, conversely, expanded, adding more analysis and explaining more in detail the relevance of the Authors' findings.
The paper has not a Conclusion (I mean, a proper conclusion and/or a dedicated section), and that is quite peculiar. A Conclusion is necessary and, in that, the Authors should quickly summarize their findings, explain further their research goals, and provide a synthesis of the methods and 'tools' they used to achieve them. This would help also to explain to what extent their paper is relevant in the current panorama of studies.
The language and written style are ok.
The paper is good, it just requires some enhancements, the addition of an actual Literature Review (with more works commented and analyzed), an expanded Discussion, and a proper Conclusion.
I indicate the need of a 'major revision' just because I think that the possible change in format would require some work from the Authors, otherwise it could be also a 'minor revision', without any problem (at all).
Thank you very much.
Round 2
Reviewer 2 Report
All ok, the paper was already good, and the Authors have improved it.
Well done!